# Long-Term Outcome in Adult Patients with Drug-Resistant Epilepsy Submitted to Vagus Nerve Stimulation

**DOI:** 10.3390/brainsci14070639

**Published:** 2024-06-26

**Authors:** Samuele Santi, Filomena Fuggetta, Gabriella Colicchio, Manuela D’Ercole, Alessandro Izzo, Quintino Giorgio D’Alessandris, Benedetta Burattini, Renata Martinelli, Nicola Montano

**Affiliations:** 1Department of Neuroscience, Neurosurgery Section, Università Cattolica del Sacro Cuore, 00168 Rome, Italy; samuele.santi01@icatt.it (S.S.); colicchiogabriella@gmail.com (G.C.); quintinogiorgio.dalessandris@policlinicogemelli.it (Q.G.D.); benedetta.burattini@gmail.com (B.B.); renata.martinelli01@icatt.it (R.M.); 2Department of Neurosurgery, Fondazione Policlinico Universitario Agostino Gemelli IRCCS, 00168 Rome, Italy; mariafilomena.fuggetta@policlinicogemelli.it (F.F.); manuela.dercole@policlinicogemelli.it (M.D.); alessandro.izzo@policlinicogemelli.it (A.I.)

**Keywords:** drug-resistant epilepsy, vagus nerve stimulation, long-term outcome, epilepsy surgery

## Abstract

Epilepsy treatment primarily involves antiseizure medications (ASMs) to eliminate seizures and improve the quality of life, but many patients develop drug-resistant epilepsy (DRE), necessitating alternative interventions. This study aimed to evaluate the long-term efficacy and safety of vagus nerve stimulation (VNS) in managing DRE. We retrospectively analyzed data from 105 adult patients treated at Agostino Gemelli Hospital from 1994 to 2022. Among the 73 patients with follow-up data, 80.8% were responders, experiencing significant reductions in seizure frequency over an average follow-up period of 9.4 years. Although 19.2% were non-responders, many of these patients still opted for generator replacements due to improvements in quality of life, such as fewer falls and shorter post-ictal periods. The overall complication rate was 12.3%, with most complications being mild and manageable. These findings suggest that VNS offers substantial long-term benefits for patients with DRE, improving seizure control and quality of life. This study underscores the importance of VNS as a viable long-term treatment option for DRE, highlighting its potential to significantly enhance patient outcomes and quality of life.

## 1. Introduction

The first-line treatment of epilepsy relies on the utilization of antiseizure medications (ASMs) that have the main aim of eliminating seizures and improve the overall quality of life of these patients. Nonetheless, a big issue in the medical management of epilepsy is that ASMs can fail to reach the goal of achieving freedom from seizures. Moreover, ASMs can be associated with different side effects, especially when more drugs are required to control the seizures. It is well known that multi-drug therapy for epilepsy can be associated with the onset or worsening of psychiatric and metabolic disorders requiring additional medications to treat these conditions. Further, it has been evidenced how the probability of being seizure-free is not increased by adding new ASMs to the medical therapy [1]. Thus, the failure of two appropriately chosen and tolerated ASMs (whether as monotherapies or in combination) to achieve sustained seizure freedom when used for an adequate period of time is the actual definition of “drug-resistant epilepsy” (DRE) [2]. DRE is associated with different problems such as depression and anxiety, cognitive impairment, difficulties in maintaining social relations and increased risk of morbidity and mortality. When a diagnosis of DRE is established, a surgical approach is advised. Epilepsy surgery has a long history and well-described and standardized techniques that permit the efficient and safe treatment of this condition. Surgery for DRE can be conceptually divided into resective surgery and palliative surgery. The aim of resective surgery is to obtain a patient who is seizure-free without postoperative neurological deficit. Classical indications for resective surgery are the following: epilepsy associated with structural lesions, such as tumors, vascular lesions, temporal lobe epilepsy due to hippocampal sclerosis, glioneuronal tumors and type-2 focal cortical dysplasia. In cases of DRE patients with no detectable abnormalities on brain magnetic resonance imaging (MRI), the concept of the so-called epileptogenic zone (EZ) is of paramount importance. The EZ has been defined as the site of onset and primary organization of epileptic seizures [3] but it has been recently redefined as the minimum amount of cortex that must be resected (inactivated or completely disconnected) in order to cure the seizures [4]. Unfortunately, despite the use of non-invasive and invasive diagnostic, such as video-EEG recording, MRI with a specific protocol for epilepsy [5], functional MRI, interictal FDG-PET, Ictal SPECT, neuropsychological evaluation and stereo-electro-encephalography, there is a significant amount of patients in whom it is impossible to identify a single EZ due to the lack of concordance between the clinical, anatomical and electrophysiological data. In these cases, a palliative surgical strategy is advised. Conceptually, the aim of palliative surgery, in contrast to the resective approach, is to obtain a decrease in seizure frequency and an overall better quality of life. There are different palliative surgical procedures that can be used to achieve this goal such as corpus callosotomy, multiple subpial transections, vagus nerve stimulation (VNS), deep brain stimulation, and responsive neurostimulation with different indications and results [6].

VNS is a palliative therapeutic strategy for patients with DRE who are not candidates for ablative surgery or who have not responded optimally to such a treatment [2]. As mentioned above, “drug-resistant epilepsy” refers to a medical condition in which substantial relief from seizures is not achieved despite the administration of appropriate antiepileptic medications at well-tolerated dosages [7]. VNS is mainly indicated for epilepsy with multiple independent and bilateral foci, for symptomatic generalized epilepsy with diffuse epileptogenic abnormalities, and for refractory idiopathic generalized epilepsy. It is also indicated for drug-resistant depression [8]. The stimulation involved is neurostimulation, where electrical pulsatile waves are directed towards the existing nervous tissue with the aim of influencing epileptic discharges in the brain [9]. This method of stimulation induces symptomatic effects that translate into therapeutic benefits [9,10]. Although initial applications date back to the 1980s [11], it took some time for the method to establish itself in the treatment of refractory epilepsy. However, the underlying operating principle of VNS remains somewhat elusive, as the exact mechanism of action of VNS in epilepsy control is far from being completely elucidated [6]. 

The current implantable treatment device comprises a compact battery-powered stimulator necessitating battery replacement roughly every six years [12]. This device is equipped with a fine wire electrode that extends from it and is typically wound around the left cervical vagus [13]. However, in certain circumstances, where accessing the left vagus nerve is deemed impractical, there is the possibility of using the right vagus nerve [12]. Notably, due to the right vagus’s innervation of the sinoatrial node, any stimulation on this side is preferably conducted under ECG monitoring [14].

The initiation process of the device typically involves administering short-lived stimulation to the vagus nerve, lasting anywhere from 30 to 90 s [11]. While the stimulation parameters exhibit considerable diversity, they commonly encompass a frequency range of 20 to 50 Hz and a pulse duration of up to 500 microseconds (with a range extending from 130 to 500 microseconds), alongside an activation period ranging from 30 to 120 s, followed by a rest period of 5 min [11,12]. It is important to note that the intensity of stimulation typically diminishes over time [11]. Furthermore, there is a clear relationship between increasing the current intensity and the enhancement of neurotransmitter release, such as norepinephrine, as well as an elevation in the firing rate of cells within the locus coeruleus [15].

A solid grasp of the fundamental principles of nerve conductance and firing thresholds is imperative in the context of VNS. The vagus nerve comprises a complex network of nerves, each characterized by its unique threshold for activation. Primarily composed of small, unmyelinated C fibers, with a smaller fraction being myelinated A and B fibers, this neural ensemble exhibits varying thresholds for stimulation. A-beta fibers, which possess the lowest firing threshold, are initially activated, while C fibers require higher current intensities for stimulation [16]. However, direct nerve stimulation may lead to discomfort, which could impede intensity escalation. Moreover, the interplay between the current intensity and pulse width requires careful consideration. Broadening the pulse width while maintaining equal current intensities facilitates heightened VNS effects [17].

Some research suggests that the vagus nerve plays a pivotal role in suppressing the kindling of seizures in regions susceptible to heightened excitability, such as the limbic system, thalamus, and thalamocortical projections [18]. Various hypotheses suggest that VNS interferes with synchronized patterns of brain activity associated with seizure activity. It is believed that VNS may affect multiple pathways, including sensory pathways and the nucleus tractus solitarius (NTS), leading to increased activity in thalamocortical projection pathways and decreased activity in areas such as the amygdala and hippocampus [19]. Additionally, VNS may influence neurotransmitter systems, particularly norepinephrine and serotonin, which could increase seizure threshold through enhancing the activity of the locus coeruleus and raphe nuclei [20,21]. Other potential mechanisms include the modulation of regional cerebral blood flow, alteration of neurotransmitter concentrations, and activation of anti-inflammatory pathways [22,23].

In studies concerning the VNS technique, the usual outcome indicator is a reduction in seizure frequency by more than 50%. While this metric effectively gauges treatment efficacy in the short term, it fails to capture the long-term evolution of epilepsy and the associated improvements in patients’ quality of life over the years. In a recent literature review on this topic, it has been evidenced how the majority of published studies have a short follow-up (FU) [6].

This study endeavors to address this gap by presenting a case series of VNS implantations in adult patients with very-long-term FU. It aims to elucidate how treatment response is sustained over time through the evaluation of ongoing reductions in seizure frequency and the potential enhancements in patients’ quality of life, even in cases where the treatment does not meet the threshold for full responsiveness. Further, we discuss our results, taking into account the pertinent literature.

## 2. Materials and Methods

A comprehensive retrospective study was carried out, encompassing a cohort of 105 adult patients, all aged 18 years and above, who underwent implantation of a VNS device at the Agostino Gemelli Hospital. The study period spanned from March 1994 to December 2022.

Patients’ demographic characteristics, including age, sex, and the etiology of seizures, were documented (Table 1). Clinical outcome was determined by comparing the ‘monthly’ seizure frequency after stimulation at last evaluation with the seizure frequency during the 3-month pre-implantation period (baseline) by using the following equation: (seizures/month on VNS − baseline seizures/month)/(baseline seizures/month) × 100. Patients were considered “responders” if they experienced a seizure frequency reduction of 50% or more with respect to baseline.

As outcome indicators, we evaluated the rate of responder patients (patients with a seizure frequency reduction of more than 50%), and in order to assess the broader impact of VNS therapy, the frequency and the reasons for battery device replacements were also recorded. Further, the number and the type of complications were also assessed.

## 3. Results

This study included a cohort of 105 patients spanning from 1994 to 2022, of whom 7 patients (6.7%) were deceased due to unrelated causes, and 25 patients (23.8%) were lost during clinical FU, leaving 73 patients actively engaged in this study (see Table 2).

### 3.1. Responder Patients

Out of the 73 patients, 59 (80.8%) were classified as responders to VNS. The mean FU in this group was 9.4 years (±5.3) with a minimum FU duration of 1 year and a maximum of 20 years. Of them, 35 patients (59.3%) were males and 24 patients (40.6%) were females, with a mean age of 33.4 years (±12.4). Among the responders, 33 patients (55.9%) underwent generator replacement, while 26 patients (44.1%) have not yet required a device replacement. Noteworthy, 13 patients in the responder group (22%) had an FU longer than 10 years and 6 responder patients (10.2%) showed an FU longer than 15 years (Figure 1).

### 3.2. Non-Responder Patients

In our series, we categorized 14 out of 73 patients (19.2%) as non-responders to VNS. In this group, there were six males (42.9%) and eight females (57.1%), with a mean age of 30.4 years (±9.6). The mean FU in the non-responder group was 9.4 years (±6.0), with a minimum FU duration of 1 year and a maximum of 21 years.

Remarkably, in this group, eight patients (42.9%) underwent generator replacement, exhibiting a discernible improvement in their quality of life (Table 3). Among the non-responder patients, three cases (21.4%) had an FU longer than 10 years and another three patients (21.4%) had an FU exceeding 15 years (Figure 2).

Furthermore, we were able to document the rate and the type of complications encountered by patients during FU. We had a total rate of complications of 12.3%, all of which were deemed non-life threatening. These included one case of device infection, one wound dehiscence, two lead breakages, four cases of hoarseness persisting for more than 1 month but less than 6 months, and one occurrence of gastroesophageal reflux.

## 4. Discussion

The present study aimed to elucidate the clinical outcomes and long-term efficacy of VNS as a palliative treatment for DRE in adult patients, based on a retrospective analysis spanning 28 years. In the literature, there are numerous studies confirming the effectiveness of VNS as a palliative strategy for treating DRE. However, data focusing on the long-term outcomes of such patients are not frequently reported.

A notable strength of this study lies in its prolonged FU period, both in responder and non-responder groups, ranging from a minimum of 1 year to a maximum of 21 years. Overall, 25 patients (34.24%) in this series had an FU longer than 10 years, supporting the evidence of the long-term effectiveness of VNS in managing drug-resistant epilepsy. Comparatively, the existing literature predominantly reports studies with a shorter FU, often not exceeding two years [20,24,25]. A recent meta-analysis by Batson et al. (2022) evaluated four clinical trials and six observational studies, reporting a mean FU duration of only 28.8 months (±26.6) [26], with only one study reporting a maximum FU of 13 years [27].

Our study’s analysis underscores the remarkable outcomes observed in responders, with an impressive achievement of 80.8%. The current literature reports a response rate of VNS of around 42% with a mean FU of 17.35 months [6]. The result of this case series is likely attributed to the extensive FU period (see Table 4). This finding resonates with the existing literature, as indicated by Englot et al., who reported a persistent challenge in achieving seizure freedom among individuals with DRE undergoing VNS therapy, but the increased FU time allowed them to see a significant increase in the achievement of this result [28]. They demonstrated that following VNS implantation, the seizure frequency saw an average reduction of 45%, with a further decrease of 36% observed at 3–12 months post-surgery, and a substantial 51% reduction after more than one year of therapy. Notably, at the last FU, approximately 50% of the patients experienced a reduction in seizures by 50% or more, with VNS correlating to a ≥50% reduction in seizures, underlining its efficacy [28]. It has been evidenced how the natural history of epilepsy is largely unknown due to the extensive use of ASMs and that, among patients who experience a remission, many remain seizure-free after ASM withdrawal, suggesting that the underlying seizure-generating factor has remitted [29]. Thus, we could speculate that seizure reduction might have happened regardless of VNS therapy. Nonetheless, in the context of VNS therapy, Elliot et al. reported a notable mean seizure reduction of 55.8% following a mean FU period of 5 years. Impressively, a substantial proportion of the participants achieved ≥75% (40.5%) and ≥50% (63.75%) seizure control, reaffirming the therapy’s efficacy over the long term. Furthermore, the mean reduction in seizures demonstrated a consistent improvement with increasing FU duration, as evidenced by a mean decrease in the seizure frequency of 76.3% among the participants with over 10 years of FU [30,31]. Batson et al. reinforced the significance of a prolonged FU in capturing seizure freedom events. Their findings revealed seizure freedom in 15 out of 273 individuals with DRE, highlighting the importance of extended monitoring in assessing treatment outcomes comprehensively [26]. Regarding the etiology of epileptic seizures in the patients considered in the study, it is noted that they are largely comparable to the data found in the literature. A recent meta-analysis by Caccavella et al. (2022) revealed that the primary cause of epilepsy in patients undergoing VNS implantation is unknown (60.63%), followed by structural causes (24.47%) and infections (7.45%) [6]. Similarly, in our case series, cryptogenic causes were found to be the most common (56.2%), followed by anoxic–ischemic (17.8%) and structural causes (11%). Infectious causes rank fourth, comprising 4.1% of the total (Table 1). However, these data fail to explain the significant differences observed in the outcomes of our case series compared to those reported in the literature. Regarding the average age at the time of stimulator implantation, we find a result consistent with data supported by the literature. Caccavella et al. report a mean age at the time of surgery of 34.28 (±2.15) years [6]; the studies included in the meta-analysis by Batson et al. present a mean age of 32.48 years [26]; in our case series, the patients underwent VNS implantation at a mean age of 32.8 (±11.9) years, with a comparable difference between the females (32.9 years) and males (32.7 years).

Another aspect of paramount importance revealed by this study concerns the quality of life of patients undergoing VNS implantation. Our case series sheds light on a notable aspect: despite not achieving a reduction in seizure frequency exceeding 50%, a substantial 42.9% of the non-responder patients opted for generator replacement surgery. Their decision was motivated by the tangible improvements in their quality of life, as evidenced in Table 3. Focusing on the motivations within our case series, it is notable that in four out of the eight cases, the motivation was a reduction in falls, which significantly impacts the daily lives of patients with epilepsy from both physical and social perspectives. Falls are a common cause of injury and fractures among patients with epilepsy, contributing to a higher rate of hospitalization in this population compared to the general population [32]. Addressing factors contributing to falls in epilepsy management is therefore crucial for improving patient outcomes and reducing the associated healthcare burden.

Furthermore, the adverse effects of antiepileptic drug treatment on patients’ physical and cognitive well-being cannot be overlooked. ASMs, targeting the central nervous system to control seizures, may lead to a range of side effects including drowsiness, dizziness, fatigue, imbalance, depression, memory problems, and irritability [33]. These side effects can exacerbate the risk of falls and impact the overall quality of life. Therefore, comprehensive epilepsy management should involve the careful consideration of both seizure control and the management of ASM-related side effects to optimize patient outcomes and enhance their overall well-being.

The remaining reasons that led the non-responder patients to replace the generator are two cases of a shorter duration of the post-ictal period, a decrease in seizure intensity, and one case of cessation of the status epilepticus condition. All these conditions underlie a significant functional impairment in the patients’ daily lives.

The utilization of VNS in patients with DRE also offers significant benefits in enhancing cognitive function. Numerous studies have demonstrated improvements in various cognitive domains following VNS therapy. Clark et al. demonstrated that VNS enhanced short-term memory, as evidenced by improved word retention in patients stimulated after reading emotionally neutral paragraphs [34]. Another study by Sun et al. revealed enhanced working memory performance in subjects receiving VNS, indicating its positive impact on cognitive function [35]. Moreover, Chan et al. found that the cognitive benefits of VNS persisted beyond one year post-implantation, further emphasizing its enduring effects on cognitive function [36]. Beyond cognitive enhancement, VNS has also been shown to improve mood, as evidenced by studies by Chan et al. and Ryvlin et al., which reported improvements in mood and quality of life among patients undergoing VNS therapy [36,37]. Additionally, Klinkenberg et al. observed reductions in anxiety, tension, and depression levels in patients with refractory epilepsy treated with VNS, further highlighting its therapeutic potential in addressing mood disorders associated with epilepsy [38]. The cognitive benefits of VNS are observed to persist over the long term, suggesting enduring effects on cognitive function [38].

VNS therapy also enables a reduction in the concomitant use of ASMs without compromising seizure control. This reduction in drug burden is crucial for patients, as excessive medication usage may lead to decreased tolerability and ultimately reduce the likelihood of achieving seizure freedom [39]. Moreover, certain ASMs are associated with metabolic consequences that can negatively impact bone health, lipid metabolism, and gonadal steroid metabolism. By reducing the drug burden, VNS therapy may mitigate the risk of such complications, thereby improving the patient’s overall quality of life [40].

Regarding the adverse events documented in our case series, they occurred in 12.3% of the patients. The most common complication was the occurrence of hoarseness in four patients. This is historically one of the most frequent adverse events, as reported by Caccavella et al., who showed that VNS is characterized by a hoarseness/dysphonia incidence rate of 37.2% [6]. This issue is due to the stimulation of the recurrent laryngeal nerve passing close to the electrode implantation site on the vagus nerve [41]. Typically, this is a transient disturbance that tends to improve and resolve within a few months, as demonstrated by the fact that all four patients showed resolution of the side effect at six months. Among the remaining cases, we found two instances of electrode rupture, one case of device infection, and one wound dehiscence. These events fall within the realm of local events, defined as having a mild impact on the patients’ quality of life and can be resolved through simple medical interventions.

VNS, therefore, not only offers promise in reducing seizure frequency but also holds immense potential in improving various aspects of one’s quality of life, making it a valuable therapeutic option for patients with DRE and related comorbidities. These findings underscore that a reduction in seizure frequency by more than 50% may not fully capture the true efficacy of this therapeutic strategy, emphasizing the need for individualized assessments to discern improvements in patients’ quality of life.

**Table 4 brainsci-14-00639-t004:** This table reports articles that investigated the effectiveness of VNS with a mean FU more than 4 years. NR: not reported.

Study	Average Follow-Up (Months)	% Responder
Casazza M, 2006 [10]	48	24%
Elliot R E, 2011 [30]	59	63.75%
Elliot R E, 2011 [31]	125	90.8%
Hoppe C, 2013 [27]	80	63%
Gonen O M, 2014 [42]	67	NR
Morace R, 2017 [43]	72	41%
Okamura A, 2020 [44]	48.6	60%
Current study	112	80.8%

### Limitations

The retrospective nature of the data collection, along with the considerable number of patients lost to follow-up, and the absence of a formal analysis regarding potential prognostic indicators of favorable outcomes represent the primary limitations of this study. Further, in this series, we were not able to perform a systematic analysis of the quality of life, for example, the change in the seizure semiology that can affect the quality of life of these patients. Despite the aforementioned constraints, we believe that our findings offer valuable insights into elucidating the long-term efficacy and quality of life outcomes associated with VNS in the management of DRE. By acknowledging these limitations and interpreting our results within this context, we contribute to the growing body of literature aimed at enhancing our understanding of the therapeutic potential and real-world implications of VNS therapy.

## 5. Conclusions

In conclusion, our study provides valuable insights into the long-term efficacy and quality of life outcomes associated with VNS in the management of DRE. While VNS is a well-established palliative strategy for patients who do not respond adequately to other treatments, such as antiepileptic drugs and ablative surgery, the precise therapeutic mechanisms remain partly elusive. Nonetheless, our analysis of a cohort of 105 adult patients over a span of 28 years reveals compelling evidence of sustained treatment response and improvements in quality of life, even among non-responder patients.

Despite acknowledging the limitations of our study, as mentioned above, our findings highlight the enduring benefits of VNS therapy, particularly in reducing seizure frequency. Moreover, the decision of a significant proportion of non-responder patients to undergo generator replacement surgery underscores the tangible improvements in their quality of life, despite not achieving the predefined threshold for treatment responsiveness.

The findings reported in this study underscore the critical importance of extending the FU duration in future research concerning the utilization of VNS in patients with DRE. By observing how treatment outcomes evolve over time, informed by both the current study’s results and existing literature, new considerations can be made regarding the indications for this therapeutic technique. Prolonged FU periods allow for a more comprehensive assessment of the sustained efficacy and long-term effects of VNS therapy. This is particularly pertinent given the chronic and refractory nature of DRE, where treatment outcomes may evolve over months and years rather than weeks. By extending the duration of FU, researchers and clinicians can capture nuances in treatment response, identify potential predictors of long-term success or failure, and refine patient selection criteria for VNS therapy. Furthermore, an extended FU duration enables the evaluation of treatment durability and the identification of any late-onset adverse events or complications associated with VNS. This information is essential for guiding clinical decision-making and optimizing patient care, ensuring that the benefits of VNS therapy outweigh any potential risks or burdens associated with long-term treatment. Moreover, prolonged FU facilitates the assessment of patient-reported outcomes and quality of life measures, providing valuable insights into the holistic impact of VNS therapy on patients’ daily functioning, psychological well-being, and overall quality of life. Moving forward, prospective studies with larger sample sizes and longer FU durations are warranted to further elucidate the clinical benefits and optimal use of VNS in the management of DRE.

## Figures and Tables

**Figure 1 brainsci-14-00639-f001:**
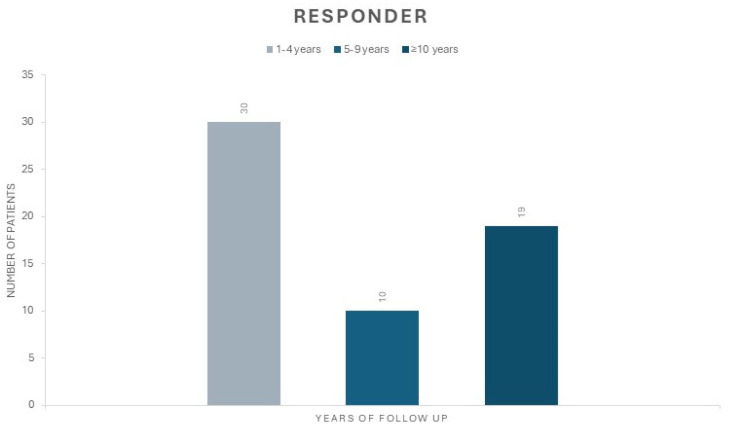
This figure shows the distribution of the number of responder patients based on FU time. The first group (grey) represents patients with a FU between 1 and 4 years, the second group (light blue) represents those with a FU between 5 and 9 years, and the third group (dark blue) represents those with an FU of more than 10 years.

**Figure 2 brainsci-14-00639-f002:**
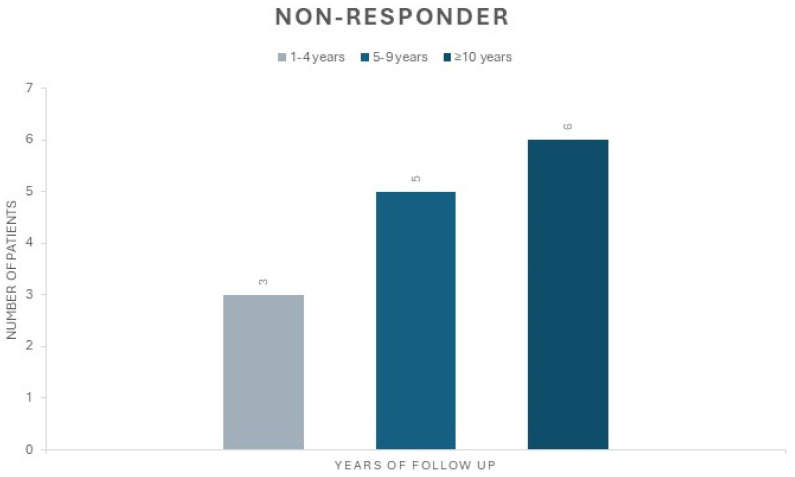
This figure shows the distribution of the number of non-responder patients based on FU. The first group (grey) represents patients with an FU between 1 and 4 years, the second group (light blue) represents those with an FU between 5 and 9 years, and the third group (dark blue) represents those with an FU more than 10 years.

**Table 1 brainsci-14-00639-t001:** This table presents an overview of the demographic characteristics of the studied population, focusing on the number of patients and their mean age and etiology.

	Male (%)	Female (%)	Total (%)
**Patients**	41 (56.2%)	32 (43.8%)	73
**Mean age**	32.7 (±11.2)	32.9 (±13.0)	32.8 (±11.9)
**Etiology**			
Cryptogenics	24 (58.5%)	17 (41.5%)	41 (56.2%)
Meningoencephalitis	2 (66.7%)	1 (33.3%)	3 (4.1%)
Anoxic–ischemic	10 (76.9%)	3 (23.1%)	13 (17.8%)
Post-tumoral	1 (33.3%)	2 (66.7%)	3 (4.1%)
Malformation—cortical dysplasia	2 (25%)	6 (75%)	8 (11.0%)
Tuberous sclerosis	0 (0%)	2 (100%)	2 (2.7%)
Neurofibromatosis 1	1 (100%)	0 (0%)	1 (1.4%)
Post-traumatic	1 (100%)	0 (0%)	1 (1.4%)
Lennox–Gastaut	0 (0%)	1 (100%)	1 (1.4%)

**Table 2 brainsci-14-00639-t002:** This table presents the difference in age, etiology, and average follow-up between the responder and non-responder group.

	Responder (%)	Non-Responder (%)
**Patients**	59 (80.8%)	14 (19.2%)
**Mean age**	33.4 (±12.4)	30.4 (±9.6)
**Etiology**		
Cryptogenics	34 (57.6%)	7 (50%)
Meningoencephalitis	2 (3.4%)	1 (7.1%)
Anoxic–ischemic	12 (20.3%)	1 (7.1%)
Post-tumoral	1 (1.7%)	2 (14.3%)
Malformation—cortical dysplasia	6 (10.2%)	2 (14.3%)
Tuberous sclerosis	1 (1.7%)	1 (7.1%)
Neurofibromatosis 1	1 (1.7%)	0 (0%)
Post-traumatic	1 (1.7%)	0 (0%)
Lennox–Gastaut	1 (1.7%)	0 (0%)
**Average follow-up (years)**	9.4 (±5.3)	9.4 (±6.0)

**Table 3 brainsci-14-00639-t003:** The table illustrates the rationale behind generator replacement for each non-responder patient.

Motivation	Patients
Decrease in the number of falls	4
Shorter duration of post-critical state	2
Lack of secondary generalization	1
Disappearance of status epilepticus	1

## Data Availability

The raw data supporting the conclusions of this article will be made available by the authors on request.

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
