# Peer review of "Long-Term Outcome in Adult Patients with Drug-Resistant Epilepsy Submitted to Vagus Nerve Stimulation"

_brainsci, 2024, doi:10.3390/brainsci14070639_

Round 1

Reviewer 1 Report

Comments and Suggestions for Authors

This study compares the seizure frequency before and after the implantation of a vagus nerve stimulator (VNS) in patients with drug-resistant epilepsy. The authors retrospectively found seizure reduction in 59 out of 73 patients in the follow-up for 1-20 years. They concluded that long-term follow-up is important for observing good outcomes of VNS therapy.

Here are my comments.
1. This study did not investigate the seizure semiology. The patient's QOL would be better when 10 GTCs/day is reduced to 5 GTCs/day than 10 GTCs/day to 5 SPS/day. We sometimes observe semiology changes after VNS implantation. It would be informative to include this point in the limitation of the study.
2. The authors emphasize the long-term follow-up to observe better outcomes of VNS, which seems to suggest that the long-term effect, rather than the short-term effect, is more impactful in the therapy. On the other hand, the seizure-generating factor would be remitted in the natural course of epilepsy in many patients (e.g., Kwan and Sander JNNP, 2004, etc). Therefore, the seizure reduction might have happened regardless of VNS therapy. It would be informative to discuss this point.
3. In Table 2, 'Lower intensity of attacks' is unclear. Also, does 'epileptic states' mean 'status epilepticus'?
4. Line 278-294: This study did not investigate the cognitive function. I think this part can be shorter, or even removed. Line 295-313: The ASM reduction was not also investigated, either.

Author Response

Response to Reviewers

Reviewer 1

1. This study did not investigate the seizure semiology. The patient's QOL would be better when 10 GTCs/day is reduced to 5 GTCs/day than 10 GTCs/day to 5 SPS/day. We sometimes observe semiology changes after VNS implantation. It would be informative to include this point in the limitation of the study.

We agree with you and added the following sentence to the limitation paragraph: “Further, in this series we were not able to perform a systematic analysis of quality of life, for example the change in the seizure semiology that can affect the quality of life of these patients”

2. The authors emphasize the long-term follow-up to observe better outcomes of VNS, which seems to suggest that the long-term effect, rather than the short-term effect, is more impactful in the therapy. On the other hand, the seizure-generating factor would be remitted in the natural course of epilepsy in many patients (e.g., Kwan and Sander JNNP, 2004, etc). Therefore, the seizure reduction might have happened regardless of VNS therapy. It would be informative to discuss this point.

 We better discussed this point in the discussion section. We also added the study you suggested and the Table 4 in which are reported the studies on VNS with a mean FU more than 4 years

3. In Table 2, 'Lower intensity of attacks' is unclear. Also, does 'epileptic states' mean 'status epilepticus'?

 We changed the Table 2 according to your suggestions. Now this Table is the Table 3

4. Line 278-294: This study did not investigate the cognitive function. I think this part can be shorter, or even removed. Line 295-313: The ASM reduction was not also investigated, either.

 We agree with you that these points are not investigated in our series. However we decided to keep these paragraphs in our discussion section to underline the positive effects of VNS reported in the literature also on these aspects

Reviewer 2

1. Introduction part is too long, and should be trimmed and more to the point. For example, from line 71 to 124, the author gives a detailed introduction to VNS. I think all these are not necessary because this is not a review paper about VNS, an introduction part author should just point out what the gap is in the current research area and what their contribution is. If the author wants to give a detailed description of how they perform VNS it can be added to the material and methods section.

 We agree with you that the introduction is long. Nonetheless we think that this paragraph is useful to give a brief overview about the technical aspects of VNS and its mechanism of action.

2. Introduction parts need more references. Many statements were made without a necessary reference. For example from line 42 to Line 55: “it is a common experience that epileptic patients usually assume more than two drugs and it is well known that there is resistance by the patient and often by the caregiver in accepting a surgical procedure to treat DRE.”

 Thank you for your comment. We decided to remove this sentence because it reflects probably our experience and not a real literature data.

3. “However, in the majority of DRE patients there are no detectable abnormalities on brain magnetic resonance imaging (MRI).” References need to be provided to support relevant statements.

 Thank you for your comment. Our sentence was incorrect. We rewrote this sentence as follow: “In cases of DRE patients with no detectable abnormalities on brain magnetic resonance imaging (MRI), it is of paramount importance the concept of the so-called epileptogenic zone (EZ)

4. Introduction paragraph starting line 125, author made the statement that “there is a dearth of literature examining the long-term outcomes of adult patients treated with this intervention for drug-resistant epilepsy”. In my opinion this statement is not 100% correct, because there’s many other studies that investigate the “long term effect” of VNS on DRE treatment, as the author already pointed out in the discussion section. It will be good if the author can add a brief review of relevant studies in introduction rather than discussion. Other studies usually have 2 years follow up while this study has an average of 9.4 years. Maybe add “very long term” to the claim so the statement would be more accurate.

 Thank you for your comment. The sentence “there is a dearth of literature examining the long-term outcomes of adult patients treated with this intervention for drug-resistant epilepsy” was incorrect so we decided to remove it.

We rewrote the introductive sentence as follows: “…a case series of VNS implantations in adult patients with very long term FU”. We added a Table 4 in which are reported the studies on VNS with a mean FU more than 4 years.

5. The biggest highlight of this study is its extremely long time of follow up, 28 years from 1994 to 2022, with mean FU 9 years, while most other studies usually be 2 years. However, the paper still lacks a comprehensive analysis of clinical data that span the long time period. For example, it would be interesting to learn what's the average year before the response, and what the average year of QOL has significant improvement? What could cause these differences if any? More analysis needs to be done to make use of the long follow up time, current version I don’t see such analysis.

 We agree with you about the lack of a systematic analysis of quality of life of these patients. This is impossible to do, due to the retrospective nature of data. We added the following sentence to the limitation paragraph: “Further, in this series we were not able to perform a systematic analysis of quality of life, for example the change in the seizure semiology that can affect the quality of life of these patients”

Minor comments:

6. More results should be visualized with tables and figures. The visualization of the paper needs improvement. Many results should be presented in the format of tables and figures. For example, it would be beneficial to plot the distribution of follow-up years for different patient groups, as described in section 3.1. Additionally, follow-up years should be included in Table 1.

Thank you for your suggestion. We added:

  • A table presenting the difference in age, etiology and average follow-up between the responder and non-responder group (now is the Table 2)
  • A figure showing the distribution of the number of responder patients based on FU time (Figure 1)
  • A figure showing the distribution of the number of non-responder patients based on FU (Figure 2)
  • A Table reporting articles that investigated the effectiveness of VNS with a mean FU more than 4 years (Table 4)

7. A lot of text in the discussion section should be moved to the results section. For instance, the detailed description of outcomes and comparison to other studies from line 220 to 254 should be placed in the results section. Conversely, the results section is currently a bit short and needs to be strengthened.

We increased the results section adding the Table 2 and the Figure 1 and Figure 2

Reviewer 3

1. This work reports clinicle outcomes and long-term efficacy of vagus nerve stimulation (VNS) as a palliative treatment for drug-resistant epilepsy in adult patients. The results are solid and clearly show some of the benefts of VNS. However, the work is not novel. Several other groups have reported similar works with less follow up time. I suggest adding a table or figure to compare the outcomes of existing similar works and show the effect of longer follow up time.

Thank you for your comment. We added a Table 4 in which are reported the studies on VNS with a mean FU more than 4 years.

Reviewer 2 Report

Comments and Suggestions for Authors

This paper is a clinical study that investigates the clinical outcomes of a cohort of 73 patients with drug-resistant epilepsy (DRE) who underwent vagus nerve stimulation (VNS). The study spans 28 years (from 1994 to 2022), with average patient follow up of 9.4 years, while most studies typically have only a 2-year follow-up. This extended duration enables the investigation of the outcomes of VNS on DRE patients at a very long term. The paper provides valuable insights into the clinical treatment of DRE with VNS. However, there are a few points that should be improved before publishing. Please see my detailed comments below: 

Major comments: 

Introduction part needs a rework. Several reasons listed below:

  1. Introduction part is too long, and should be trimmed and more to the point. For example, from line 71 to 124, the author gives a detailed introduction to VNS. I think all these are not necessary because this is not a review paper about VNS, an introduction part author should just point out what the gap is in the current research area and what their contribution is. If the author wants to give a detailed description of how they perform VNS it can be added to the material and methods section.

  1. Introduction parts need more references. Many statements were made without a necessary reference. For example from line 42 to Line 55: “it is a common experience that epileptic patients usually assume more than two drugs and it is well known that there is resistance by the patient and often by the caregiver in accepting a surgical procedure to treat DRE.” and “However, in the majority of DRE patients there are no detectable abnormalities on brain magnetic resonance imaging (MRI).” References need to be provided to support relevant statements.

  1. Introduction paragraph starting line 125, author made the statement that “there is a dearth of literature examining the long-term outcomes of adult patients treated with this intervention for drug-resistant epilepsy”. In my opinion this statement is not 100% correct, because there’s many other studies that investigate the “long term effect” of VNS on DRE treatment, as the author already pointed out in the discussion section. It will be good if the author can add a brief review of relevant studies in introduction rather than discussion. Other studies usually have 2 years follow up while this study has an average of 9.4 years. Maybe add “very long term” to the claim so the statement would be more accurate.

  1. The biggest highlight of this study is its extremely long time of follow up, 28 years from 1994 to 2022, with mean FU 9 years, while most other studies usually be 2 years. However, the paper still lacks a comprehensive analysis of clinical data that span the long time period. For example, it would be interesting to learn what's the average year before the response, and what the average year of QOL has significant improvement? What could cause these differences if any? More analysis needs to be done to make use of the long follow up time, current version I don’t see such analysis.

Minor comments:

  1. More results should be visualized with tables and figures. The visualization of the paper needs improvement. Many results should be presented in the format of tables and figures. For example, it would be beneficial to plot the distribution of follow-up years for different patient groups, as described in section 3.1. Additionally, follow-up years should be included in Table 1.

  2. A lot of text in the discussion section should be moved to the results section. For instance, the detailed description of outcomes and comparison to other studies from line 220 to 254 should be placed in the results section. Conversely, the results section is currently a bit short and needs to be strengthened.

Author Response

(The authors gave the same response as above.)

Reviewer 3 Report

Comments and Suggestions for Authors

This work reports clinicle outcomes and long-term efficacy of vagus nerve stimulation (VNS) as a palliative treatment for drug-resistant epilepsy in adult patients. The results are solid and clearly show some of the benefts of VNS. However, the work is not novel. Several other groups have reported similar works with less follow up time. I suggest adding a table or figure to compare the outcomes of existing similar works and show the effect of longer follow up time.

Author Response

(The authors gave the same response as above.)
